# Antiferroelectrics and Magnetoresistance in La_0.5_Sr_0.5_Fe_12_O_19_ Multiferroic System

**DOI:** 10.3390/ma16020492

**Published:** 2023-01-04

**Authors:** Jia-Hang Yin, Guo-Long Tan, Cong-Cong Duan

**Affiliations:** 1State Key Laboratory of Advanced Technology for Materials Synthesis and Processing, Wuhan University of Technology, Wuhan 430070, China; 2Department of Electronic Engineering, Luzhou Vocational and Technical College, Luzhou 646000, China

**Keywords:** antiferroelectrics, ferrimagnetism, dielectric constants, magnetoresistance

## Abstract

The appearance of antiferroelectrics (AFE) in the ferrimagnetism (FM) system would give birth to a new type of multiferroic candidate, which is significant to the development of novel devices for energy storage. Here we demonstrate the realization of full antiferroelectrics in a magnetic La_0.5_Sr_0.5_Fe_12_O_19_ system (AFE+FM), which also presents a strong magnetodielectric response (MD) and magnetoresistance (MR) effect. The antiferroelectric phase was achieved at room temperature by replacing 0.5 Sr^2+^ ions with 0.5 La^2+^ ions in the SrFe_12_O_19_ compound, whose phase transition temperature of ferroelectrics (FE) to antiferroelectrics was brought down from 174 °C to −141 °C, while the temperature of antiferroelectrics converting to paraelectrics (PE) shifts from 490 °C to 234 °C after the substitution. The fully separated double P-E hysteresis loops reveal the antiferroelectrics in La_0.5_Sr_0.5_Fe_12_O_19_ ceramics. The magnitude of exerting magnetic field enables us to control the generation of spin current, which induces MD and MR effects. A 1.1T magnetic field induces a large spin current of 15.6 n A in La_0.5_Sr_0.5_Fe_12_O_19_ ceramics, lifts up dielectric constants by 540%, and lowers the resistance by −89%. The magnetic performance remains as usual. The multiple functions in one single phase allow us to develop novel intelligent devices.

## 1. Introduction

In modern magnetoresistance (MR) electronic devices or spintronics, the switch of magnetic order causes big energy consumption since the induced magnetic field by alternating current causes resistance loss. It conflicts with the demand for faster, smaller, and ultra-low-power to electronic devices. Alternatively, by integrating multiferroics into these electronic devices, magnetization can be manipulated by voltage rather than electromagnet [1,2]. This results in a change in MR with greater energy efficiency and could make novel low-power tunable electronic devices more promising. Consequently, multiferroics consisting of ferromagnetic and ferroelectric orders were pushed forward to appear in front of us due to the capability of voltage straightforward switching magnetization through magnetoelectric (ME) coupling effect [3,4,5], which brings us a variety of unprecedented physical phenomena [6,7,8].

Multiferroic materials can be used to realize a new generation of multifunctional devices which integrate ferroelectricity and magnetism, that is, new magnetoelectric sensing devices, spin electronic devices, high-performance information storage devices [4,9]. The multiferroic material with the ME coupling effect uses voltage rather than current to regulate the magnetization direction and minimizes joule heat dissipation. It can fundamentally solve the problem of high energy consumption of storage devices [4]. The conventional multiferroic feature consists of ferroelectrics (FM) and antiferromagnetism (AFM) or ferromagnetism (FM) [1,4]. Extending the multiferroic type to the new one in combination with antiferroelectrics and ferromagnetism would be scientifically significant since it may generate new quantum phenomena, which could give birth to a new generation of information technology and energy storage devices [10,11]. In reality, however, this type of multiferroic material is rare [11,12]. 

The first AFE feature in the multiferroic family was observed in (Bi_1–x_ Re_x_) FeO_3_ (Re = La, Sm) thin film system, which demonstrates a hybrid phase of ferroelectrics and antiferromagnetism [12,13,14], while full multiferroic AFE feature was achieved in Dy_0.75_Gd_0.25_FeO_3_ system at 1.85K [13]. The rare earth orthoferrites RFeO_3_ is a combination of AFE and AFM. Its double P-E loops occurred at a very low temperature (1.85K), and the saturation polarization is pretty small (only 0.15 μC cm^−1^). Seeking more multiferroic candidates with better AFE and FM features would be a milestone since it may incubate not only more types of multiferroic memories but also produce new types of energy storage devices which could be charged either electrically or magnetically.

The advantage of the AFE state over FE one in multiferroic materials is that AFE capacitors can provide much higher recoverable energy density than FE counterparts due to the double hysteresis loops as well as much lower remnant polarization in the AFE phase [14]. FEs usually possess high dielectric constants, but the combined effects of large remnant polarizations and inferior electric field endurances limit the energy densities of FE materials to low values [15,16]. In AFEs, by contrast, adjacent dipoles of the same strength in the crystal structure are initially aligned in opposite directions. These initially antiparallel dipoles, however, can be forced to become parallel along the direction of a sufficiently strong external electric field to reach a FE state through an electric field-induced AFE–FE phase transition [16]. Then, once the external electric field is removed, the induced FE phase can revert back to the initial AFE state, thereby generating so-called double P-E hysteresis loops. The typically high electric fields associated with these AFE–FE phase transitions, coupled with the significant changes in polarization during the AFE–FE phase transition, enable a large amount of energy to be stored and released. Thus, AFE materials have a great advantage over FE counterparts for use in energy storage devices. The energy storage density of AFE materials is much higher than their linear dielectric and FE counterparts [17].

The traditional AFE materials, such as PbZrO_3_ [18], AgNbO_3_ [19], (Bi,Na)TiO_3_ [20], (Pb,La)(Zr,Sn,Ti)O_3_ [21], etc., are used to control and store electric energies and play a key role in mobile electronic devices, stationary power systems, hybrid electric vehicles [22,23]. These perovskites, however, can be charged electrically only. By introducing AFE order into multiferroic materials, we are able to make a new type of multiferroic capacitors that can store not only the electric energy but also the magnetic energy, or even both. This would provide us with one more additional freedom to charge capacitors and thus could somehow improve the energy storage density. In this paper, we will present a new type of multiferroic candidate of La_0.5_Sr_0.5_Fe_12_O_19_, which integrates full antiferroelectrics and strong ferrimagnetism together in one single phase. The room temperature state changes from ferroelectrics [24] to pure antiferroelectrics by replacing 0.5 Sr^2+^ ions with 0.5 La^2+^ ions in SrFe_12_O_19_. The newly conceived compound demonstrates fruitful new physical phenomena, including AFE and FM features, magnetoresistance (MR) effect, and colossal magnetic dielectric (MD) response. These combined functions may give birth to new electric devices, such as multiferroic memories which could able to be written electrically and read magnetically, as well as multiferroic capacitors which may provide more freedom to store magnetic energies in addition to conventional electrically stored energy in antiferroelectric capacitors, in this way, the multiferroic capacitors could be charged electrically or magnetically or both. The purpose of this paper is to develop a new type of multiferroic material that can be used to generate novel devices.

## 2. Experimental Methods

For the aim of realization of full antiferroelectrics in M-type hexaferrites, we substitute 0.5 Sr^2+^ ions with 0.5 La^2+^ ions in SrFe_12_O_19_ to obtain the La_0.5_Sr_0.5_Fe_12_O_19_ compound. A polymer precursor method was applied to prepare La_0.5_Sr_0.5_Fe_12_O_19_ nanometer powders first. The preparation details have been described in the other literature [24]. The powders were pressed into pellets with a diameter of 6.2 mm and a thickness of ~0.55 mm. The pellet specimen was sintered into ceramic at 1350 °C for 2 h and subsequently annealed in an oxygen atmosphere at 800 °C for three times, each one was maintained for 3 h while keeping the surface in the direction upward or downward. Afterward, the ceramic specimen was coated with silver electrodes on both surface sides, which was solidified at 820 °C for 15 min. The structure of the specimens was detected by an X-ray diffraction (XRD, Empyrean, Shanghai Scientific Instruments Co, Shanghai, China) machine with Cu Ka radiation. The magnetic property was measured by a Quantum Design physical property measurement system (PPMS, Quantum Scientific Instruments Trading (Beijing) Co, Beijing, China). The P-E hysteresis loop was measured on a Sawyer-Tower circuit-based ferroelectric measurement system. The temperature-dependent dielectric property and impedance spectrum were measured on a Micro test PRECISION LCR meter (6365, Suzhou Quasi-Test Instrument Technology Co, Suzhou, China), which are frequency ranges from 10 Hz to 10 M Hz. The magnetoelectric polarization performance and magnetic dielectric response were measured on a Keithley 2450 source meter as well as the same LCR instrument by applying a variable DC magnetic field on the samples.

## 3. Results and Discussion

### 3.1. Structure Identification

The X-ray diffraction (XRD) pattern reveals that the La_0.5_Sr_0.5_Fe_12_O_19_ system shares the same crystal structure with the magnetoplumbite-5H system, which may be termed as PbFe_12_O_19_ or SrFe_12_O_19_. Figure 1a displays the XRD pattern of sintered La_0.5_Sr_0.5_Fe_12_O_19_ ceramics. The discrete blue lines in Figure 1b are the standard diffraction spectrum of PbFe_12_O_19_ (PDF#41-1373). Obviously, the overall diffraction peaks of La_0.5_Sr_0.5_Fe_12_O_19_ matches well with the standard diffraction lines, indicating that the crystal structure of the La_0.5_Sr_0.5_Fe_12_O_19_ system was the same as that of the magnetoplumbite-5H model in PbFe_12_O_19_ or SrFe_12_O_19_. There is a small lattice contraction after the substitution. The substitution of 0.5 La with 0.5 Sr won’t change the lattice volume too much since those La atoms would fit into large void spaces where the Sr atoms occupied before substitution in the unit cell of SrFe_12_O_19_ lattice [24]. The decrease in the radius of La atoms compared to Sr ones is ignorable to the size of the void space. Therefore, the lattice parameters reduce very little after the replacement of La with Sr. The diffraction pattern is clean and was not indexed to second-phase impurities. However, several small peaks may be traced to the contribution of a small number of excessive Fe_2_O_3_ in the pattern. This result suggests that a pure solid solution of La_0.5_Sr_0.5_Fe_12_O_19_ in a single phase has been successfully fabricated after the substitution of 0.5 Sr^2+^ ions with 0.5 La^2+^ ions in the SrFe_12_O_19_ compound. According to this formula (La_0.5_Sr_0.5_Fe_12_O_19_), La ions would be bivalent (La^2+^) instead of trivalent in charge. The outer shells of La atoms give the electronic structure of 5d16S2, two electrons at the 6S level and one electron at the 5d level. Therefore, La atoms may lose either two outer shell 5S electrons to become La^2+^ ions in divalent or lose all three outer shell electrons (2 from 6S+1 from 5d) to be La^3+^ ions. Consequently, the valence charges of 0.5La+0.5Sr would be +2, whereas 12 Fe^3+^ ions provide +36 positive charges; the total positive charges are then counted to be +38, which may balance the total negative charges of −38 from 19 O_2-_ ions in La_0.5_Sr_0.5_Fe_12_O_19_. The charge balance in this formula would make this compound possible and may leave no vacancies in the lattice.

Figure 2 shows the microstructure of La_0.5_Sr_0.5_Fe_12_O_19_ ceramics measured by scanning electron microscopy (SEM, JSM-7500F, Japan Electronics Corporation (JEOL), Akishima, Japan), which shows that it has good denseness and most of the grains show a plate-like morphology. However, the grains of La_0.5_Sr_0.5_Fe_12_O_19_ are mostly irregular hexagonal plate-like structures, which is due to the substitution of Sr by La, which leads to the disruption of the hexagonal symmetry of the crystal structure of SrFe_12_O_19_. The [0001] direction is perpendicular to the surface of the plate grains. The plate shape indicates that the grains do not grow preferentially along the [0001] direction but preferentially along the [1000] and [0100] directions. Obviously, the grains also grow at different rates along the [1000] and [0100] directions. Therefore, the grains do not have a perfect hexahedral shape but actually grow into irregular plates.

### 3.2. Dielectric Relaxing and Antiferroelectrics of La_0.5_Sr_0.5_BaFe_12_O_19_ Ceramics

The temperature-dependent dielectric spectrum of pure SrFe_12_O_19_ ceramics has been reported in the literature [25]. That dielectric spectrum demonstrated two-phase transition peaks, one for ferroelectrics to antiferroelectrics at 174 °C (TF-A) and the other one for antiferroelectrics to paraelectrics at 490 °C (TA-P) [25], suggesting that SrFe_12_O_19_ is a ferroelectric phase at room temperature. Thereafter, by replacing 0.2 Sr ions with 0.7 La ions in SrFe_12_O_19_, the first phase transition peak (TF-A) shifts to the vicinity of room temperature, where the ceramic specimen shows a partial antiferroelectric feature coexisting with ferroelectric state [26]. Herein, the substitution concentration of La ions is further increased to 0.5, aiming to move the first TF-A peak down to low-temperature region. As such full antiferroelectric state may be achieved at room temperature. Figure 3 exhibits the temperature-dependent dielectric loss spectrum of La_0.5_Sr_0.5_Fe_12_O_19_ ceramics as a function of frequency. There also appear two-phase transition peaks. One (TF-A) locates at the low-temperature region, representing the transformation of ferroelectrics (FE) to antiferroelectrics (AFE); the other one (TA-P) stands at high-temperature region corresponding to the transition of antiferroelectrics to paraelectrics (PE). The first TF-A peak spans from −158 °C to −141 °C when the frequency increases from 50 kHz to 200 kHz, and the second TA-P peak reversely shifts from 250 °C down to 234 °C as the frequency increases from 50 kHz to 200 kHz. Both transition peaks demonstrate strong relaxing behavior upon frequencies. However, the second TA-P peak relaxes reversely to the first one due to the antiferroelectric dipolar feature. It can be seen that the ferroelectric state can only appear at the temperature region below −141 °C, while the paraelectric phase may occur above 234 °C (for 200 kHz). There is a wide space between −141 °C to 234 °C, in which antiferroelectrics remains in a thermodynamically stable state.

Upon the substitution of 0.5 Sr with 0.5 La in SrFe_12_O_19_, the first transition peak (TF-A) was successfully moved from 174 °C down to a low-temperature region below −140 °C, while the second TA-P peaks shifted from 490 °C down to 234 °C, leaving a broad space (−141 °C ~234 °C) at the vicinity of room temperature to accommodate full antiferroelectric state. In this way, a full antiferroelectric state at room temperature was successfully achieved, as shown in Figure 3.

The spectrum of the real part of the dielectric constant reveals similar changes in the phase configuration of the La_0.5_Sr_0.5_Fe_12_O_19_ system. The temperature-dependent dielectric constant $\left(\varepsilon_{r}^{\prime }\right)$ spectrum is displayed in Figure 4. The transition peak of TF-A (FE→AFE) locates at −145 °C, while that of TA-P (AFE→PE) is centered at 232 °C (200 kHz). The critical temperature of TF-A in La_0.5_Sr_0.5_Fe_12_O_19_ system moved to a very low temperature far below room temperature (RT) when 50% Sr was replaced with 50% La, leaving a wide space of −145 °C ~ 232 °C for the antiferroelectric state to survive as a stable thermodynamic phase. Consequently, the RT state of La_0.5_Sr_0.5_Fe_12_O_19_ changes to a full antiferroelectric phase after the rare earth elemental substitution. Both real and imaginary parts of dielectric constants varying with temperature revealed similar phase configurations, which was demonstrated in Figure 4. However, there is a small retardation between the two types of spectra. The real part lags behind the imaginary one by about 18 °C because of the phase difference between the real part and the imaginary part.

Figure 5 displays a double P-E hysteresis loop as the experimental evidence for proving the pure antiferroelectric feature of La_0.5_Sr_0.5_Fe_12_O_19_ ceramics. The polarizations in two hysteresis loops align in opposite directions upon switching the electric field. In AFEs, adjacent dipoles of the same strength in the crystal structure are initially aligned in opposite directions leading to zero overall spontaneous polarization. These initially antiparallel dipoles, however, can be forced to become parallel along the direction of a sufficiently strong external electric field to reach a FE state [16]. Then, once the external electric field is removed, the induced FE phase can revert back to the initial AFE state, thereby generating so-called double P-E hysteresis loops. The FE loops are separated by a linear AFE component, whose polarization counteracts each other due to the antiparallel dipoles. The better the linearity. The purer is the antiferroelectric feature. It may be seen from Figure 4 that the linear AFE component is almost overlapping with the x-axis, and the net polarization approaches zero through the cancellation of reversely aligned dipoles in the full antiferroelectric phase of the La_0.5_Sr_0.5_Fe_12_O_19_ system. Only when the external electric field is sufficiently high enough to exceed ±, EA may the antiparallel dipoles be forced to align along the same direction with the E field and thus generates the appearance of double P-E hysteresis loops. The maximum saturated polarization is estimated to be 45 μC cm^−2^ at E = 235 kV cm^−1^ and the remnant polarization is around 3.3 μC cm^−2^, almost approaching zero. The forward switching (AFE-to-FE) field EA locates at ±109 kV cm^−1^, backward switching (FE-to-AFE) field EF is around ±131 kV cm^−1^, the switching hysteresis is $\Delta E=E_{F}-E_{A}=22\;\mathrm{kV}\;{\mathrm{cm}}^{-1}$.

The two switching fields (EA & EF) of the La_0.5_Sr_0.5_Fe_12_O_19_ system are comparable to that of classical antiferroelectric perovskites, such as Pb (Zr_x_Ti_1−x_) O_4_ [27,28] and AgNbO_3_ [29] whose EA ranges from 40 kV cm^−1^ to 170 kV cm^−1^ and EF ranges from 100 kV cm^−1^ to 230 kV cm^−1^. The recoverable storage energy density of the La_0.5_Sr_0.5_Fe_12_O_19_ system is calculated from the hysteresis loops in Figure 5 to be 6.2 J cm^−3^ at 235 kV cm^−1^. This value is also similar to that of PLZT perovskite capacitors (6.7 J cm^−3^) [30]. Higher energy density could be only achieved in a 700 nm La-doped PbZrO_3_ thin film system by applying a very high field of up to 600 kV cm^−1^ [31] or a 500 nm Sr-doped PbZrO_3_ thin film with an even higher field of 900 kV cm^−1^ [32].

### 3.3. Magnetic Property of the La_0.5_Sr_0.5_Fe_12_O_19_ Ceramics

Strong magnetism is an important indexing property to identify the candidate as a multiferroic compound in addition to its ferroelectrics. The SrFe_12_O_19_ ceramic performances in ferrimagnetism with high remnant magnetic moments and large coercive field. The replacement of 0.5 Sr^2+^ with 0.5 La^2+^ may not alternate its magnetic properties too much. Figure 6 shows an M-H hysteresis loop of La_0.5_Sr_0.5_Fe_12_O_19_ ceramic, which was measured upon a Physical Property Measurement System (PPMS). The specimen was sintered at 1350 °C and subsequently annealed in an O_2_ atmosphere at 800 °C for 6 h. The remnant moment (M r) of the La_0.5_Sr_0.5_Fe_12_O_19_ ceramic specimen is around 29.5 emu g^−1^, and coercive force (H c) is about 4020 O e. By comparing these parameters with that of the SrFe_12_O_19_ ceramic specimen sintered at 1150 °C, we found that the coercive force was enhanced from 3510 O e to 4020 O e after the replacement of 50% Sr with 50% La in the SrFe_12_O_19_ system, suggesting that the magnetism was improved somehow. The heavy substitution by rare earth elemental La ions generates a new type of multiferroic candidate, whose molecule formula could be termed La_0.5_Sr_0.5_Fe_12_O_19_. This novel multiferroic compound combines full antiferroelectrics and ferrimagnetism together in one single phase at room temperature.

### 3.4. Magnetoelectric Coupling Effect in La_0.5_Sr_0.5_Fe_12_O_19_ System

The combination of FE (or AFE) and FM states in one single phase is the fundamental element of multiferroics. The magnetoelectric (ME) coupling effect, on the other hand, is the prerequisite requirement for multiferroics to be practicable. The application needs magnetically read-out and electrically write-in operation in multiple state memories. Here we will present the ME coupling performances in La_0.5_Sr_0.5_Fe_12_O_19_ ceramics. The ME coupling was tested by a setup system, and the variation of spin current and capacitor of the specimen with the external magnetic field was measured using a Keithley 2450 Source meter or Micro test LCR meter. The change in the physical parameters of the La_0.5_Sr_0.5_Fe_12_O_19_ ceramic specimen upon different magnetic fields was recorded by these instruments and output to a computer. La_0.5_ Sr_0.5_ Fe_12_O_19_ ferrite crystallizes in a hexagonal structure with 64 ions present on each unit cell, located in 11 different symmetry sites (P63/mmc space group). The crystal structure is shown in Figure 7. 24 Fe^3+^ ions are located in five different sites: including three octahedral sites (12k, 2a, and 4f2), one tetrahedral lattice site (4f1), and one hexahedral lattice site (triangular biconical lattice site) (2b) The magnetoelectric coupling in La_0.5_ Sr_0.5_ Fe_12_O_19_ is generated by three parallel (2a, 12k and 2b) and two anti-parallel (4f1 and 4f2) sublattices through the superexchange interaction between O^2−^ ions. The magnetoelectric coupling-induced polarization arises from the inverse Dzyaloshinskii-Moriya-type antisymmetric effect of spin-orbit coupling [33], where two unequal obliquely tuned spin pairs are coupled, and two nonconjugated spins with a generic nonparallel configuration interact with each other. As it is in other ferrites of SmFeO_3_ and DyFeO_3_ [34], the coupling between these two helical spins generates spin currents that are generating spin currents, which in turn, induce electric polarization. So, the current measured by the Keysight 2450 source meter is the spin current.

We believe that a spin current is formed in our system. Figure 8 displays the variation of ME spin currents and the real part of dielectric constants of La_0.5_Sr_0.5_Fe_12_O_19_ ceramic specimen with increasing magnetic field. The Schottky barrier is formed between the silver electrode and the specimen surface, and the coupling voltage has no effect on the barrier height of electrons on the silver electrode side. However, the coupling voltage has a great influence on the barrier height of electrons on the specimen side, and the larger the applied voltage, the lower the barrier height. When the magnetic field is small, the resulting coupling voltage is also called smaller, and the barrier height of electrons on the silver electrode side is smaller, and the electrons flowing from the silver electrode to the sample dominate at this time, and the coupling current is negative at this time. As the magnetic field increases, the resulting coupling voltage also increases. Accordingly, the potential barrier of electrons at the silver electrode side is not affected by the voltage and remains unchanged. The height of the potential barrier of electrons at the specimen side decreases as the voltage increases. At this time, the electron flow from the specimen to the silver electrode increases continuously. At 700 mT, the two-electron flows roughly cancel, and the current is close to 0. Continue to increase the magnetic field and the electron flow from the specimen to the silver electrode. The current is positive at this time.

It can be seen from Figure 8a that the initial spin current drops with a magnetic field below 200 m T, above which the ME coupling becomes strong enough that the spin current starts to increase with a magnetic field. The ME coupling current origins from the interaction of two neighboring conic spins, whose ${\vec{k}}_{0}\times \left(S_{i}\times S_{j}\right)\neq 0$ gives rise to spin currents upon the external magnetic field. Within the region of 0~200 m T, the magnetic structure in the La_0.5_Sr_0.5_Fe_12_O_19_ system may arrange the conic spins to a downward orientation. Larger magnetic field strength will force more conic spins to align downwards, which is antiparallel to the direction of the magnetic field. In this region, thus, the ME coupling current becomes negative and decreases with the strength of the magnetic field. In the region of 200 m T < B < 800 m T, however, the magnetic structure changes to be the intermediate I phase, where the axis of conic spins will align to an upward direction with a large inclination angle to the z-axis (direction of magnetic field). The interaction of two such conic spins generates ME coupling current with the same inclination upward direction. The component on the z-axis of these spin currents slowly increases when the magnetic field becomes stronger, while those components in the other directions will cancel each other due to the random distribution of orientations. Therefore, the spin current within this region exhibits a slow increment with a magnetic field. When the magnetic field exceeds 800 m T, the magnetic profile switches to the intermediate II phase, where the axis of conic spins aligns with a small inclination angle to the z-axis. As the magnetic field increases, more and more conic spins will follow up the magnetic field to line upwards, aligning themselves parallel to the direction of the magnetic field. As such, the ME coupling current increases sharply with the magnetic field, as shown in Figure 8a.

Figure 8b shows the change in the real part of dielectric constants (ε′) with a magnetic field. It can be seen that the dielectric response to the magnetic field is highly consistent with the ME coupling spin current. The stronger the magnetic field. The larger the dielectric constant. The increment of constant dielectric marches in the same step with the spin current. Above 800 m T, ε′ increases exponentially with the strength of the magnetic field. A higher magnetic field produces a larger spin current, which generates larger magnetically induced electric polarization (*P*(*H*)) since $P\left(H\right)=\int_{0}^{\infty }idt$ (where i is the spin current, and t is the coupling time). There is a relationship between the increment of ε′ and *P*(*H*), which is expressed as follows [25]:(1)$$\begin{array}{cc} \varepsilon_{r}\left(E,H\right) & =\frac{0.036\pi t\times P\left(E,H\right)}{U}\cdot \frac{1}{1+\omega^{2}\tau^{2}}\;\\ & =\frac{0.036\pi t\times \;\left[P\left(E\right)+P\left(H\right)\right]}{U}\cdot \frac{1}{1+\omega^{2}\tau^{2}}\;\\ & =\varepsilon_{r}\left(E\right)+\varepsilon_{r}\left(H\right)\; \end{array}$$
where *t* is the thickness, *ω* is the frequency, *τ* is relaxation time, *U* is the voltage, *P(E)* is the *E* field-induced polarization, and *P(H)* is the magnetic field-induced polarization. When *H* = 0, *ε_r_(E, H)* = *ε_r_*(*E*). Upon the application of an external magnetic field, an additional item of *εr*(*H*) as a function of *P*(*H*) will subjoin the item to *ε_r_*(*E*), enhancing the dielectric constant to be $\varepsilon_{r}\left(E,H\right)=\varepsilon_{r}\left(E\right)+\varepsilon_{r}\left(H\right)$. Higher B induces larger *P*(*H*), which lifts up more *ε_r_*(*E,H*) accordingly.

Figure 9 shows the shift of spin current waves with an external magnetic field as a function of coupling time, within the first magnetic structure region of B < 200 m T, the current wave downward shifts from −0.7 n A to −1.6 n A when the magnetic field increases from 50 m T to 200 m T due to the downward orientation of the conic spins. Within the second profile of the intermediate, I phase (200 m T < B < 800 m T), the ME current change very slowly with the magnetic field, resulting in the overlapping of the spin current waves. When the magnetic field is larger than 800 m T, the magnetic structure goes into the profile of the intermediate II phase, where the spin current enhances largely with the magnetic field. This shift pattern provides a better-visualized view to observe the change of spin current with magnetic field in La_0.5_Sr_0.5_Fe_12_O_19_ ceramics.

Figure 10A shows the variation of complex impedance upon the magnetic field as a function of frequency. In the low-frequency region (f < 100 Hz), the impedance (Z) reduces largely with a magnetic field. Within the high-frequency region (f > 1 kHz), the change in Z is much smaller. The magnetic field strength increases from 0mT to 1100mT, and the impedance spectrum drops from 28.3 kΩ to 14.7 kΩ, exhibiting a strong MR effect. Figure 10B shows the stepwise change of impedance or resistance with the magnetic field at 50 Hz (a) and MR change ratio at different stepwise magnetic fields (b). Again, the impedance shows an increment with the magnetic field at the first magnetic profile (0 < B < 200 m T), then it drops gradually with B within the second profile of the intermediate I phase (200 m T < B < 800 m T). Above 800 m T, Z reduces sharply with B. The MR change ratio in Figure 10B exhibits a reversal stepwise variation process. MR change ratio is calculated using Equation ΔZ = [Z(B) − Z(0))/Z(B)]%. The enhancement stepwise of the MR ratio above 800 m T is much larger than that below this critical field. The stronger the magnetic field, the larger the ratio. The maximum MR change ratio is around −89% at 1100 mT. The size of the MR effect that happens in this ferrite oxide is comparable to that appears in Fe/Cr/Fe trilayers film system [35,36] as well as Cu/Co multilayers [37]. Obviously, the changing profile in impedance or resistance with magnetic field reversely matches the change mode in ME spin current. Both profiles follow up with the evolution process of the magnetic structure. Therefore, the orientation of the conic spins within different magnetic structure regions determines the profile of the magnetoelectric coupling effect or MR profile.

## 4. Conclusions

The main novelty of this work is that we developed a new type of multiferroic candidate with a formula of La_0.5_Sr_0.5_Fe_12_O_19_, which combines antiferroelectrics and ferrimagnetism together. The La_0.5_Sr_0.5_Fe_12_O_19_ ceramics were fabricated by replacing 0.5 Sr ions with 0.5 La ions, forming a solid solution in a single phase. The pure antiferroelectric feature of La_0.5_Sr_0.5_Fe_12_O_19_ ceramics was proved by double electric polarization (P-E) hysteresis loops being fully separated by a linear AFE component with zero net polarization. In the AFE part, adjacent dipoles of the same strength in the crystal structure are initially aligned in opposite directions leading to a zero overall spontaneous polarization. The first phase transition peak for FE to AFE locates at −141 °C, while the second transition peak for AFE to PE is positioned at 234 °C. The antiferroelectric state remains a thermodynamically stable phase at a wide space of −141 °C to 234 °C. The room temperature falls in the scope of this range. Therefore La_0.5_Sr_0.5_Fe_12_O_19_ ceramic behaves in full antiferroelectrics at RT. No doubt, La_0.5_Sr_0.5_Fe_12_O_19_ ceramic also performs in strong ferrimagnetism. Meanwhile, this compound demonstrates a strong magnetoelectric coupling effect and magnetoresistance (MR) effect. The magnetic field (B = 1.1T) induced a large spin current up to 15.6 n A, uplifting the dielectric constant by more than 500%. The impedance was reduced by 89% upon a B = 1.1T field, exhibiting an MR effect in a bulk oxide. The change in dielectric constant and impedance with the magnetic field matches well with the magnetic profile, which is actually controlled by the orientation of conic spins.

## Figures and Tables

**Figure 1 materials-16-00492-f001:**
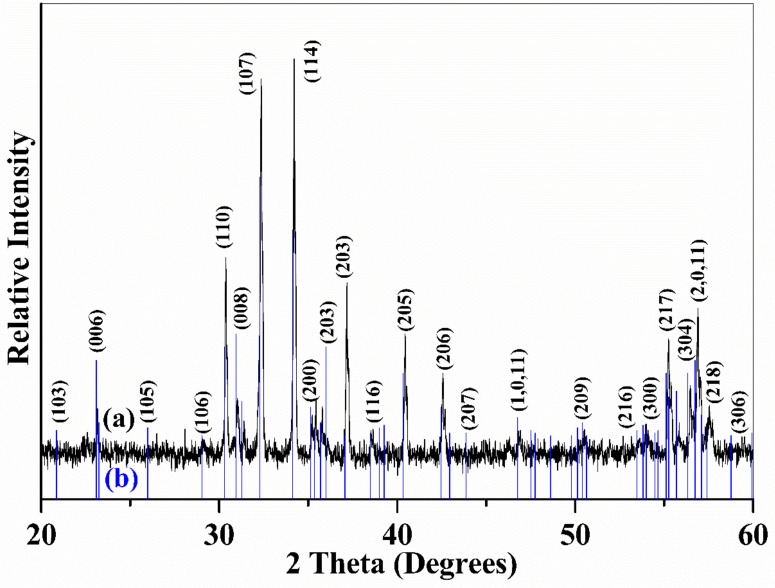
(**a**), XRD pattern of La_0.5_Sr_0.5_Fe_12_O_19_ ceramic, which was sintered at 1350 °C for 3 h; (**b**) the underneath discrete blue lines are standard diffraction spectrum of PbFe_12_O_19_ (PDF# 41-1373).

**Figure 2 materials-16-00492-f002:**
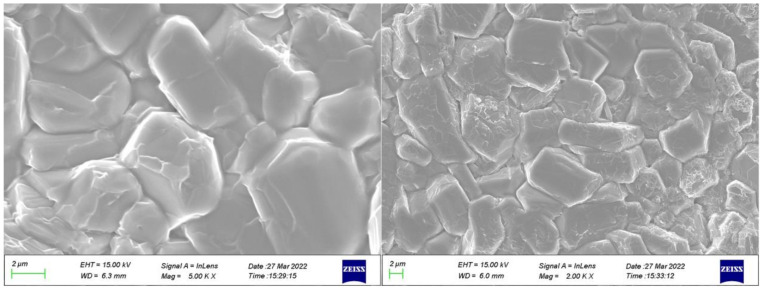
SEM images of La_0.5_Sr_0.5_Fe_12_O_19_ ceramic, which were sintered at 1350 °C for 3 h.

**Figure 3 materials-16-00492-f003:**
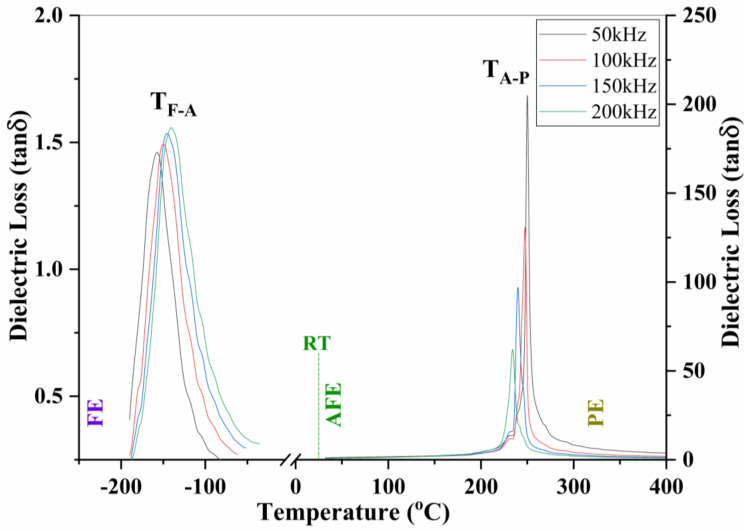
The temperature-dependent dielectric loss spectrum of La_0.5_Sr_0.5_Fe_12_O_19_ ceramics as a function of frequency within the range of −200 °C to 400 °C, and the frequency spans from 50 kHz to 200 kHz. RT represents room temperature.

**Figure 4 materials-16-00492-f004:**
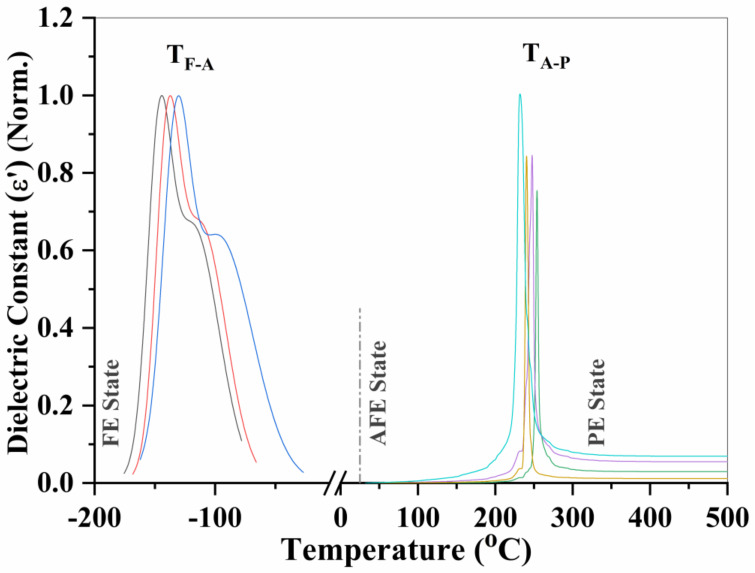
The temperature-dependent dielectric constant spectrum of La_0.5_Sr_0.5_Fe_12_O_19_ ceramics (real part ε’) with frequency spanning from 50 kHz at left to 200 kHz at right for TF-A peaks, while it diffuses reversely for TA-P peaks. All the datasets have been normalized so as to be comparable with each other.

**Figure 5 materials-16-00492-f005:**
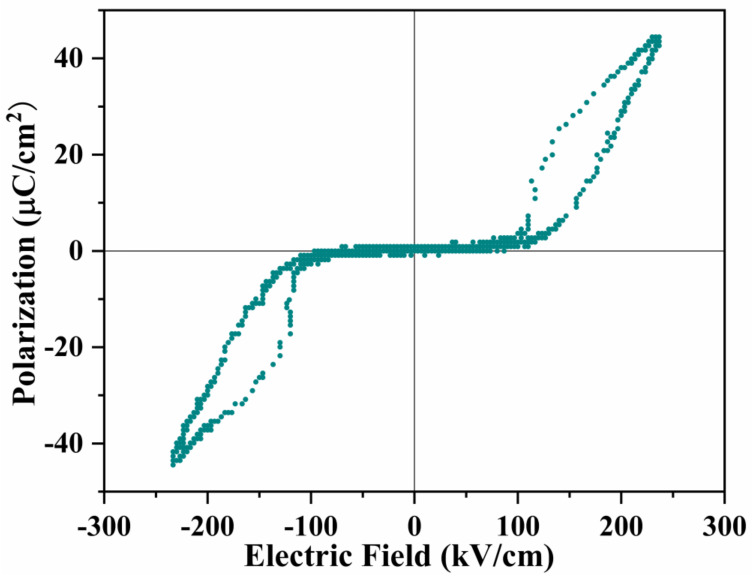
Fully saturated double P-E hysteresis loops as the experimental evidence for the antiferroelectrics of the La_0.5_Sr_0.5_Fe_12_O_19_ ceramics.

**Figure 6 materials-16-00492-f006:**
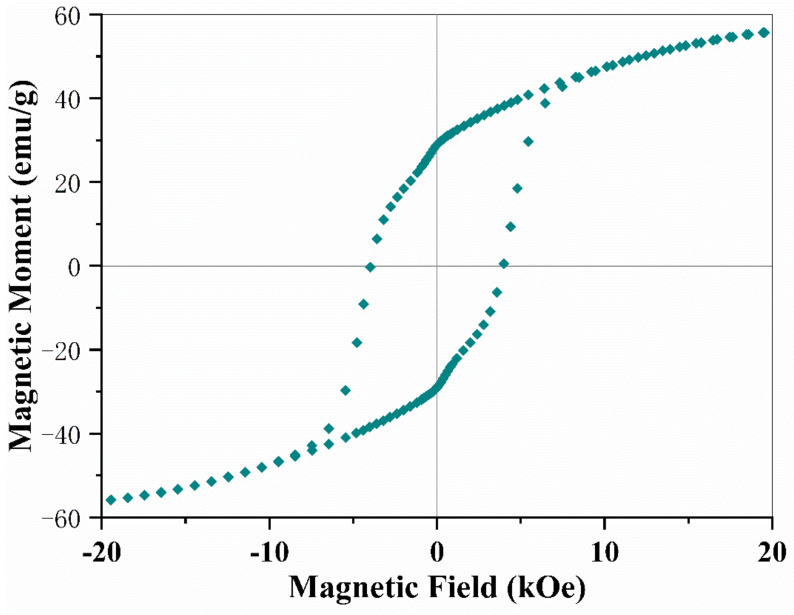
The magnetic M-H hysteresis loop of La_0.5_Sr_0.5_Fe_12_O_19_ ceramics, being sintered at 1350 °C for 3 h and subsequently annealed in an O^2^ atmosphere at 800 °C for three times, each one was maintained 3 h.

**Figure 7 materials-16-00492-f007:**
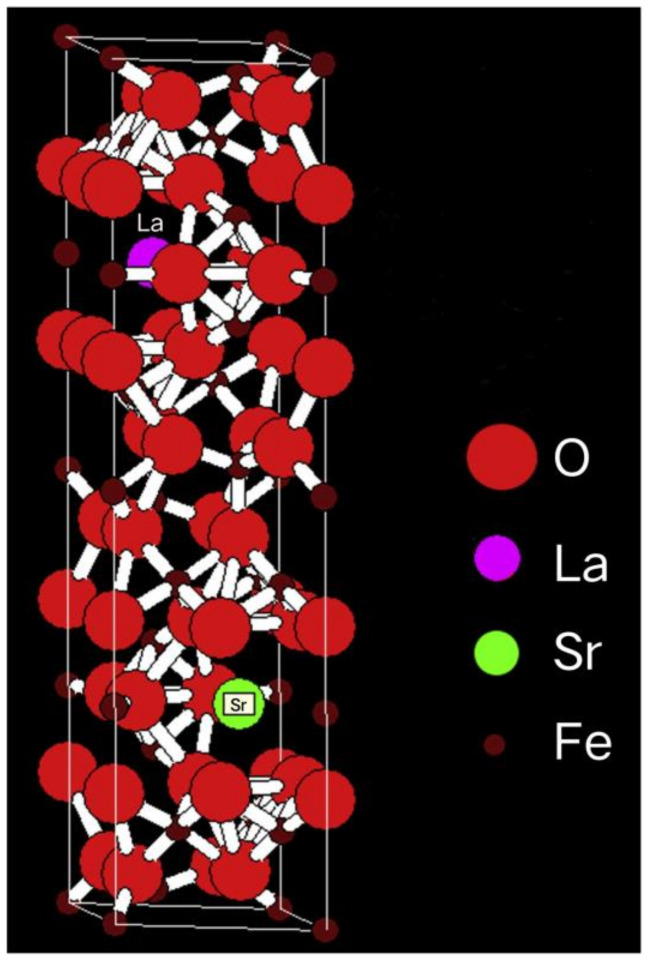
Schematic crystal structure of La_0.5_Sr_0.5_Fe_12_O_19_ with space group P63/mmc.

**Figure 8 materials-16-00492-f008:**
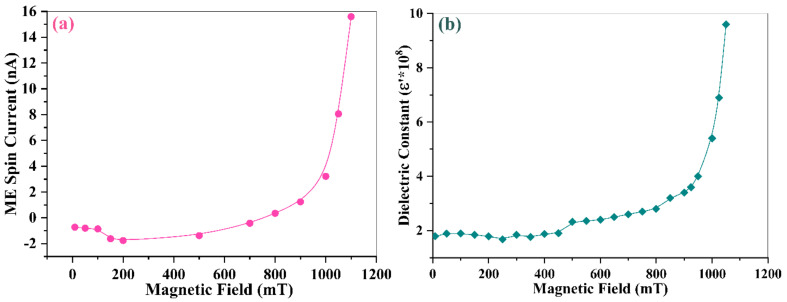
The magnetoelectric coupling performance of La_0.5_Sr_0.5_Fe_12_O_19_ ceramic specimen: (**a**) variation of spin current with magnetic field and (**b**) change in dielectric constant upon different magnetic fields at 50 Hz.

**Figure 9 materials-16-00492-f009:**
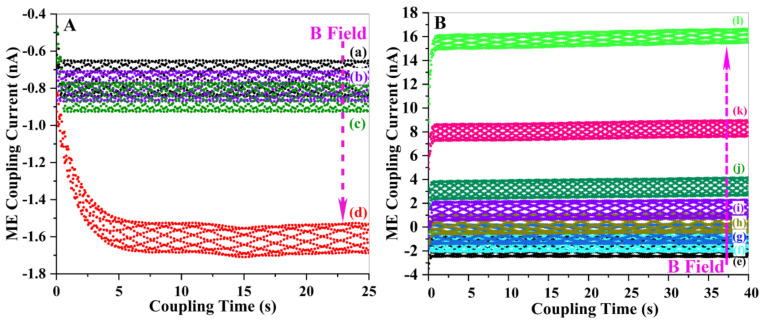
The shift of spin current waves with the external magnetic field in La_0.5_Sr_0.5_Fe_12_O_19_ ceramics; (**A**): (a) 50 m T, (b) 100 m T, (c) 150 m T, (d) 200 m T, (**B**): (e) 400 m T, (f) 500 m T, (g) 600 m T, (h) 700 m T, (i) 800 m T, (j) 900 m T, (k) 1000 m T, (l) 1100 m T.

**Figure 10 materials-16-00492-f010:**
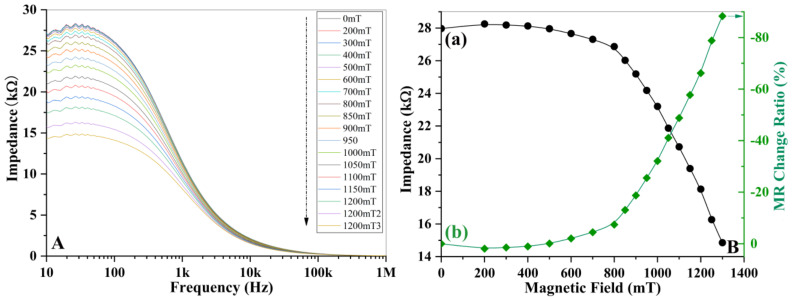
Magnetoresistance effect (MR) in La_0.5_Sr_0.5_Fe_12_O_19_ ceramics; (**A**): Change in impedance or resistance with the magnetic field as a function of frequency, (**B**): reduction process of the impedance with the magnetic field (a) and MR change ratio upon different magnetic fields (b). The magnetic field increases from 0mT to 1200 m T at a step of 50 m T, and the magnetic field stays at each stepwise where a full frequency dielectric spectrum was collected using the scanning mode.

## Data Availability

Not applicable.

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
