# Peer review of "Antiferroelectrics and Magnetoresistance in La0.5Sr0.5Fe12O19 Multiferroic System"

_materials, 2023, doi:10.3390/ma16020492_

Round 1
Reviewer 1 Report
In the manuscript entitled "Antiferroelectrics and Giant Magnetoresistance in La0.5Sr0.5Fe12O19 Multiferroic System", the authors demonstrated the realization of full antiferroelectrics in magnetic La0.5Sr0.5Fe12O19 system (AFE+FM), which also presents strong magnetodielectric response (MD) and giant magnetoresistance (GMR) effect. Besides The antiferroelectric phase was achieved at room temperature by replacing 0.5 Sr2+ ions with 0.5 La2+ ions in SrFe12O19 compound, whose phase transition temperature of ferroelectrics (FE) to antiferroelectrics was brought down from 174 ºC to -141ºC, while the temperature of antiferroelectrics converting to paraelectrics (PE) shifts from 490ºC to 234ºC after the substitution. The fully separated double P-E hysteresis loops reveal the antiferroelectrics in La0.5Sr0.5Fe12O19 ceramics. The magnitude of exerting magnetic field enables us to control over the generation of spin current, which induces MD and GMR effects. A 1.1T magnetic field induces large spin current of 15.6 nA in La0.5Sr0.5Fe12O19 ceramics, lifts up dielectric constants by 540% and lowers the resistance by -89%. The manuscript provides detailed information, and the validity and usefulness of the proposed method were demonstrated through a series of experiments. The manuscript is within the scope of the journal. The work is very interesting and, therefore, I recommend the manuscript for publication in "Materials" after minor revision.
1) I didn't see the Graphical Abstract figure. Did the authors make it?
2) In the last paragraph of the introduction the authors state: "In this paper, we will present a new type of multiferroic candidate of La0.5Sr0.5Fe12O19, which integrates full antiferroelectrics and strong ferromagnetism...", "The newly conceived compound demonstrates fruitful new physical phenomena, including AFE and FM features, giant magnetoresistance (GMR) effect and colossal magnetic dielectric (MD) response." Why only this composition (La0.5Sr0.5Fe12O19)? Did the authors test other compositions between La and Fe? If yes, what were the results? It would be very interesting to have this justification in the article. I understand that here the authors tried to highlight the objective of the work, but the novelty of the work is not expressed. In which aspect this work is original and better than others?
3) Section 2. Experimental Methods is very short. Please enter more details (technical parameters) of the techniques used for characterization.
4) In the manuscript, the authors report analyzes carried out on the La0.5Sr0.5Fe12O19 system, with very interesting multiferroic properties for science and industry. However, I would like to ask: (a) how many samples were used in this study? (b) how many syntheses?
5) Did the authors make any microscopy measurements to visualize the samples? SEM or TEM (especially the latter would be very good for comparing XRD data)?
6) Some figures need to have better resolution.
7) Figure 1 appears before being cited in the text. The same occurs with figures 7 and 8.
8) Less than 50% of the references are recent (2017-2022), which is inadmissible. Please replace the old references with newer ones.
9) Authors should check if the references are in accordance with the journal's norms.
10) There are some writing style and grammatical errors. Some sentences in the manuscript are incorrectly written with ambiguous meanings. I recommend that these errors should be corrected by a native English speaker, or a reviewer software.
Reviewer 2 Report
The considered manuscript presents the results of investigation of La-Sr hexaferrite. Some new interesting experimental date are considered and discussed.
But there are some remarks to the text which should be answered.
The expression "giant magnetoresistance" and corresponding abbreviation "GMR" are used for the effect which observed in the layered structures (magnetic-nonmagnetic-magnetic) nowdays. For the subject of the manuscript this names can't be used. I recommed to change them with "magnetoresistance" and "MR". There is no the equation for magnetoresistance calculation, so the date of figure 8b (green) have no physical sense.
The hexafirrites have some magnetic sublattice so they must be called "ferrimagnetics" not "ferromagnetics". The correction though the text must be made.
For explanation of the discussed effects it would be useful to add the figure with crystal cell atoms position.
For the line 100 the corresponding references must be presented.
In the caption to figure 1(b) , the word "green" should be changed with "blue" (lines 119 and 123).
I have a doubt that coercivity was really measured with accuracy till unit of Oersted (lines 241-242). I recommend to write about two singnificant digits. I suspect that magnetic viscosity was not taken into account.
The authors present the result of measurement of the spin current (line 256, figure 6a). But there is no the description of this parameter measurements. All devices (lines 257,258) measure the current, not spin current. So the statement about spin current measurements must be proved. In my opinion, authors are talking about ordinary electric current, not spin.
The date of the figure 6a present both negative and positive current. It looks like schottky effect, but not discussed in the text.
The impedance oscillations with frequency (figure 9a) look like experimental noise. The explanation or discussion should be presented.
Round 2
Reviewer 2 Report
The revised manuscript is significantly improved compared to the original version. The authors accepted, or responded in detail to all the comments. Nevertheless, there are some typos and not entirely conclusive statements.
When describing experimental methods, it is said about annealing samples three times for three hours. However, the caption to Figure 6 says that the annealing will last 6 hours. Adjustment is required.
Line 131 should probably have the word "substitution", but not "substation".
In line 242, the exponent needs to be raised.
In the answers to reviewer, the authors note that the accuracy of measuring the magnetic field reaches 1%, nevertheless, they give 6 significant digits for the coercive force. In line 258, it is nesessary to write "from 3540 Oe to 4020 Oe".
Since the review is not a place for scientific discussion, I ask the authors to indicate "we believe that a spin current is formed in our system", for example, in line 294. Although I draw the attention of the authors that in this case there should be two subsystems, in one of which spins with one sign are accumulated, and in the second - with another.
